# Evidence of Limitations of the Transconductance-to-Drain-Current Method ($g_m/I_d$) for Transistor Sizing in 28 nm UTBB FD-SOI Transistors

**Leonardo Barboni**

Electrical Engineering Institute-FING-UDELAR, Julio Herrera y Reissig 565, Montevideo CP 11300, Uruguay; lbarboni@fing.edu.uy

**Abstract:** The transconductance-to-drain-current method is a transistor sizing methodology that is commonly used in CMOS technology. In this study, we explored by means of simulations, a case of study and three figures of merit used for the method, and we conclude for the first time that the method should be reformulated. The study has been performed on Ultra-Thin Body and Buried Fully Depleted Silicon-On-Insulator 28 nm low-voltage-threshold NFET commercial technology (UTBB FD-SOI), and the simulations were performed via Spectre Circuit Simulator, by using the device model-card. To our knowledge, no previous attempts have been made to assess the method capability, and we collected very important results that infer that the method should be reformulated or considered incomplete for use with this technology, which has an impact and ramifications on the field of process modeling, simulation and circuit design.

**Keywords:** UTBB FD-SOI 28 nm; transconductance to-drain-current transistor sizing methodology limitations; CAD tools and methodologies for low-power

## 1. Introduction

The transconductance-to-drain-current method (widely known also as the $g_m/I_d$ method) is a broadly used tool for calculating CMOS transistor dimensions starting from specifications at the analog circuit (or system) level. Its broad usage and effectiveness are because it provides a unified sizing procedure for CMOS transistors for analog low-power and radiofrequency applications (from the strong to weak inversion regions). It has proven to be a method that is accurate and technology independent. It uses the curve of the transconductance ($g_m$) over the bias drain current ($I_d$) ratio $g_m/I_d$ relationship (also named transconductor efficiency) versus the normalized bias current $I_d/(W/L)$ ($W$ and $L$ are the transistor width and length respectively). The set of curves $g_m/I_d$ generated for a particular technology are continuous for the various regions of operation. The method has been well explained and detailed in [1–3].

Past studies have demonstrated the effectiveness of this sizing methodology. It is accurate, is technology-independent and can be applied to more advanced CMOS technology, allowing the designer to implement current-independent sizing designs. The method is also adapted and can be used for a short channel length of $L = 180$ nm [4,5] and below [6]. Additionally, it is essential to emphasize that in the literature, it has been reported that many circuit designs based on the $g_m/I_d$ design flow have achieved the desired specifications (it has been verified by simulations and measurements). All used design flows have the common factor of being based on width-independent parameters that we describe in the next section, and which are the core characteristics of the method that we assessed in this study.

Currently, technology downscaling is a consolidated process in the semiconductor industry that has triggered considerable interest regarding addressing novel analog circuit design methods. That is motivated by reported studies that have argued regarding the difficult or non-viability of scaling CMOS technologies to analog designs (for instance in [7]). However, the Ultra-Thin Body and Buried Oxide Fully-Depleted Silicon-on-Insulator (UTBB FD-SOI) has been reported as a promising technology to continue the scaling of silicon metal-oxide semiconductor field-effect transistors [8]. In this framework, the question that arises is whether the $g_m/I_d$ method is still useful in the case of UTBB FD-SOI 28 nm industrial technology. Alternatively, better stated: is the method complete and does it have the capability to handle changes in technology that involve extreme downscaling? This manuscript aims to address this question and provide early conclusions and suggestions for the circuit designers' community.

In the recent reference [9] (Appendix 3) is explored the layout dependence of the $g_m/I_d$ design method for the 65 nm-CMOS technology. The mentioned conclusion is that designers must decide on a suitable number of finger width and number of fingers to assume the $g_m/I_d$ design method to be the right design approach. Our work reinforces such considerations about layout dependent effects on the device under study. As the deep-scaled FD-SOI devices usage becomes widespread to continue CMOS scaling, our work heads up to designers who indeed will find an inconsistent result between the performance of their built circuits by means of the transconductance-to-drain-current theoretical approach and the performance obtained by simulations using the technology model-card given by the vendor (or foundry), that is assumed to be reliable. The ramifications of this work are important.

## 2. Compound Transistor Principle for the $g_m/I_d$ Method

The $g_m/I_d$ method (Section 2 of [3]) is based on the assumption that the curve of $g_m/I_d$ versus $I_N$ (normalized bias current $I_N = I_d/(W/L)$) is independent of the MOS transistor size (or formally speaking, the $W/L$ factor) and drain current biasing. The main conclusion, among others, is that $g_m/I_d$ depends on $g_m/g_{ds}$ which is width-independent. Let us provide better insight into this relation by explaining the superposition principle. For this we reproduced the compound transistor scheme that is used as the key hypothesis to support the method (Figure 1). To explain the DC characteristics of the arrangements shown in Figure 1, for all cases, we used the same $V_G$, $V_D$, $V_{BPn}$, $W$ and $L$ values for UTBB FD-SOI transistors (assumed to be ideal and without mismatched geometry). Therefore, case (b) is a compound of transistors of case (a), whereas case (c) is a single one but has a doubled width, $W$. For parallel transistors, case (b) can be treated as one merged transistor with a width of 2 W. Because the SOI transistor equations predict a bias current value that is proportional to $W$ (ref. [10,11] and also because the PDK model of our technology uses charge-surface potentials, where the threshold voltage $V_{TH}$ does not depend on $W$), it is easy to conclude that all transistor cases are biased on the same $g_m/I_d$ (see the detailed explanation in [5]).

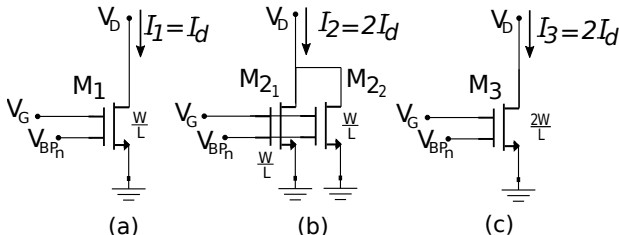

**Figure 1.** Scheme of compound UTBB FD-SOI transistors biased at the same $g_m/I_d$ value, where $V_G$ represents the gate voltage, $V_D$ is the drain voltage and $V_{BPn}$ is the back-gate bias voltage, $I_1$, $I_2$, $I_3$ represent the drain bias current for each considered case, and $W$ and $L$ represent the width and channel length, respectively.

Therefore, on the basis of the previous geometrical argumentation of the compound transistor scheme when using the $g_m/I_d$ method, and because of the transistor width ($W$) proportionality of some of their features, the following assumptions, which are the core figures of merit of the $g_m/I_d$ method, are stated:

1.  The unity gain frequency $f_T = \frac{1}{2\pi} g_m/C_{gg}$ is expected to be independent of $W$, where $C_{gg} = C_{gs} + C_{gb} + C_{gd}$ is the total gate capacitance [5].
2.  The intrinsic gain, $g_m/g_{ds}$ is expected to be independent of $W$ [5].
3.  $g_m/g_{bs}$ is expected to be independent of $W$ [5].

Of course, for these last statements to hold, we need to ensure that the parallel compound transistor principle applies.

## 3. Methodology for Preliminary Assessment of the $g_m/I_d$ Method

In this letter, rather than studying an overwhelming number of examples, we investigated UTBB FD-SOI 28 nm low-voltage-threshold nfet commercial technology, and the simulations were performed using Spectre [12] (using the tool Cadence [12]) with the vendor-provided PSP model-card that also uses intrinsec parasitic capacitances). This technology has a minimum transistor size of 30 nm gate length ($L$) and 80 nm gate width ($W$). First, utilizing simulations, we reviewed the compound transistor principle validity. We selected $L = 180$ nm for the study presented here because this gate channel length has been used and reported to be effective for CMOS circuits designs using the $g_m/I_d$ method in many studies (for instance in [5]). Consequently, we maintained this transistor channel length value in our study for the sake of completeness and to better compare with other results. Subsequently, to verify if the figures of merit maintain their width-independence properties, as mentioned above, we obtained the following figures of merit via simulations to graph them versus $W$: $f_T = \frac{1}{2\pi} g_m/C_{gg}$, $g_m/g_{ds}$ and $g_m/g_{bs}$. This was performed on one transistor with a fixed length and given bias voltages, by sweeping the gate-width using the following values:

*   case (1) $W \in [0.08, 0.1, 0.15, 0.2, 0.25, 0.5, 0.75, 1, 5, 10] \, \mu m$, finger = 1 for all cases.
*   case (2) $W \in [0.08, 0.1, 0.15, 0.2, 0.25, 0.5, 0.75, 1, 5, 10] \, \mu m$ with number of fingers $n_f \in [1, 1, 1, 2, 2, 5, 5, 10, 10, 20]$ respectively.

In the remaining parts of the manuscript, we discuss the simulations in the considered case of this study to assess the feasibility of this method. We mention in advance that no promising results were found.

## 4. Verification of the Compound Transistor Principle

The first test was to simulate the scenario of compound FD-SOI transistors biased at the same $g_m/I_d$, as depicted in Figure 1. For each case, we simulated a transistor size of $W = 80$ nm and $L = 180$ nm with $V_G = 0.5$ V, $V_D = 1$ V and $V_{BPn} = 0$ V (for case (c) we use used a double width). The simulated bias current result is as follows: $I_1 = 8.4328 \, \mu A$, $I_2 = 16.866 \, \mu A$ and $I_3 = 16.4079 \, \mu A$. However, for $M_1$ the resulting simulations showed: $g_m/I_d = 13.8311 \, V^{-1}$, for $M_{2_1}$ and $M_{2_2}$ the obtained result was $g_m/I_d = 13.8311 \, V^{-1}$ and for $M_3$ we obtained $g_m/I_d = 13.8331 \, V^{-1}$.

Moreover, for $M_1$ we obtained $g_m = 116.636 \, \mu\Omega^{-1}$. For $M_{2_1}$ and $M_{2_2}$ we obtained $g_m = 116.636 \, \mu\Omega^{-1}$ and for $M_3$ was $g_m = 226.973 \, \mu\Omega^{-1}$. We observed that there was a weak or reduced numerical difference between the $I_2$ value and $I_3$ (i.e., $I_2 = 2I_1$ but $I_2 - I_3 = 0.458 \, \mu A$).

We found that this result was not strong enough to argue that the presumption on which the original $g_m/I_d$ method is based is no longer valid. Hence, we continued exploring the behavior of the three figures of merit to obtain more explicit evidence.

## 5. Results and Discussion

In this section, we describe three singles benchmark tests used to ensure the applicability of the $g_m/I_d$ method. In particular, we were interested in verifying the width-independence of the figures of merit mentioned above. Simulations were rigorously verified and we analyzed the static behavior without the back-bias effect (we set $V_{BPn} = 0$) to closely mimic the n-type CMOS transistor conditions and geometries that have been reported for using the $g_m/I_d$ method. We then describe the following studies:

1.  Figure 2 shows the results obtained of a simulation for $L = L_{min} = 30$ nm, the width $W$ swept in a range $[0.08, 0.1, 0.15, 0.2, 0.25, 0.5, 0.75, 1, 5, 10]$ µm (finger = 1 for all cases) and the bias voltages set to: $V_{DD} = 1$ V, $V_G = 0.5$ V and $V_{PBn} = 0$ V. The differences between the extreme values of the figures of merit are: $[f_t]_{max} - [f_t]_{min} = 8.8$ GHz, $[g_m/g_{ds}]_{max} - [g_m/g_{ds}]_{min} = 6.01$ and $[g_m/g_{bs}]_{max} - [g_m/g_{bs}]_{min} = 0.48$.
2.  Figure 3 shows the results obtained of a simulation for $L = 180$ nm, the width $W$ swept in a range $[0.08, 0.1, 0.15, 0.2, 0.25, 0.5, 0.75, 1, 5, 10]$ µm (finger = 1 for all cases) and the bias voltages values set to: $V_{DD} = 1$ V, $V_G = 0.5$ V and $V_{PBn} = 0$ V. The differences between the extreme values of the figures of merit are: $[f_t]_{max} - [f_t]_{min} = 2.5$ GHz, $[g_m/g_{ds}]_{max} - [g_m/g_{ds}]_{min} = 381.3$ and $[g_m/g_{bs}]_{max} - [g_m/g_{bs}]_{min} = 0.49$.
3.  Figure 4 shows the results for the simulation with $L = 180$ nm, the total width $W$ swept in a range $[0.08, 0.1, 0.15, 0.2, 0.25, 0.5, 0.75, 1, 5, 10]$ µm with finger $n_f = [1, 1, 1, 2, 2, 5, 5, 10, 10, 20]$ respectively (the total $W$ is split up). The bias voltages set to: $V_{DD} = 1$ V, $V_G = 0.5$ V and $V_{PBn} = 0$ V. The differences between the extreme values of the figures of merit are: $[f_t]_{max} - [f_t]_{min} = 3.6$ GHz, $[g_m/g_{ds}]_{max} - [g_m/g_{ds}]_{min} = 309.7$ and $[g_m/g_{bs}]_{max} - [g_m/g_{bs}]_{min} = 0.39$.

For discussion, we first mention the critical finding of the width-dependence shown in the figures. Parameters $f_t$ and $g_m/g_{bs}$, showed reduced variations, that were below 10%. However, it could not be ensured that the design methodology used for the circuit design was affected. Formally, it is not possible to guarantee robustness concerning such a slight variation in the numerical scheme that links transistors features and circuit-level performance during the design stage. From our point of view, this is a factor to be taken into account in the integrated circuit design flow. Another exciting result arise, which indicated that more detailed studies are required. In ref. [13], the authors showed $f_T$ independence from the gate finger width in 28 nm FD-SOI devices. Conversely, in our study, we showed the effects of the gate fingers in Figure 4. It is our opinion that the technology has not achieved maturity, and variations in the device built physical processes lead to controversial results.

The most significant proof of the width-dependence of the figure of merit $g_m/g_{ds}$ is depicted in Figure 3, which showed an estimated variation above 70% and in Figure 4 (with gate fingers), with an estimated variation above 70% for the variation range of the considered $W$. The width-dependence effect is reduced when using reduced $L$ values, employing $L_{min}$.

On the other hand, from Figure 4 we observe certain plateau (or tendency to be a constant value) of the $g_m/g_{ds}$ and $g_m/g_{bs}$ parameters at the center of the figure, where the $W/n_f$ values lie into the range 0.1 µm and 0.15 µm. It witnesses the possible $g_m/I_d$ method independence concerning layout configuration, more specifically regarding the finger width but not with $W$ as primarily assumed.

By means of singles-case studies, we show that the method should be reformulated to take into account several effects that are currently not captured by the $g_m/I_d$ method, principally the $g_m/g_{ds}$ figure of merit.

Going back to Figure 1, we conclude that a device of $2W/L$ is not equivalent to two parallel devices of $W/L$ ratio. It becomes more pronounced at deep sub-micron and very wide devices. Since by modifying $W$ the extrinsic capacities values change, all extrinsic capacities have to be included in the $g_m/Id$ method. Therefore, the method may become a not treatable and challenging one, losing its desired "hand analysis" capability.

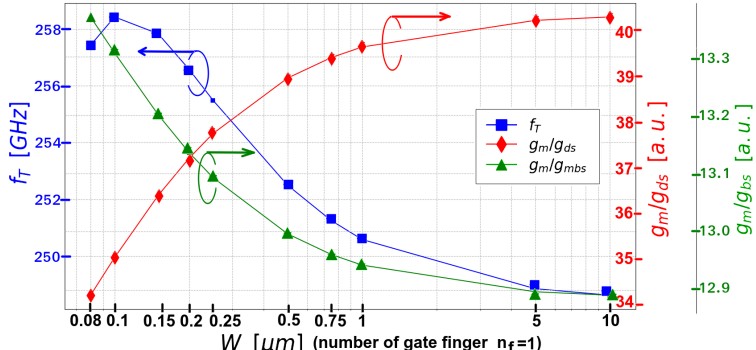

**Figure 2.** Obtained simulations for $L = L_{min} = 30$ nm, indicated bias values and $W$ sizes (finger = 1 for all cases). Please, observe that $L$ is constant for all cases, hence, it is not necessary to plot $W/L$ on the $x$-axis.

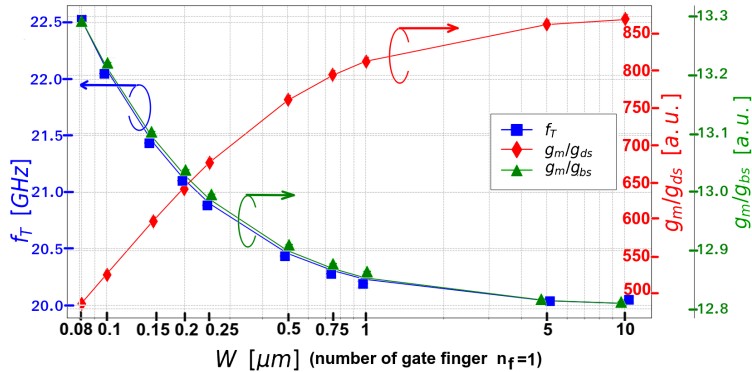

**Figure 3.** Obtained simulated for $L = 180$ nm, indicated bias values and $W$ sizes (finger = 1 for all cases). Please, observe that $L$ is constant for all cases, hence, it is not necessary to plot $W/L$ on the $x$-axis.

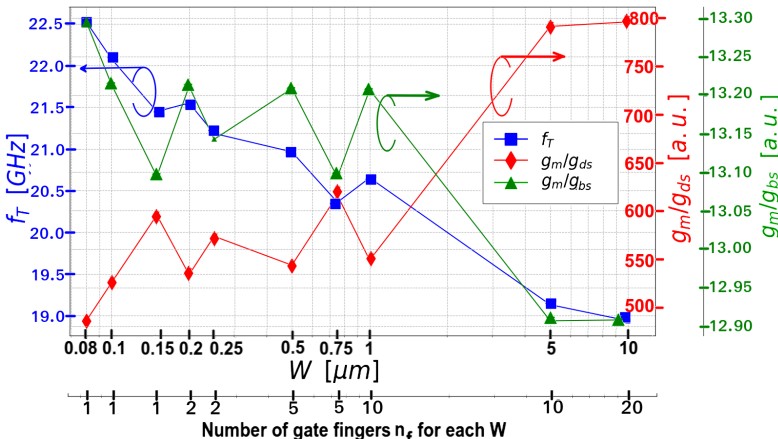

**Figure 4.** Obtained simulated parameters for $L = 180$ nm, the indicated bias values and $W$ sizes with indicated finger values. Please, observe that $L$ is constant for all cases, hence, it is not necessary to plot $W/L$ on the $x$-axis.

## 6. Example: Simple Design Case

To have a better insight into the problem treated in this work, we study the circuit shown in Figure 5: a common-source amplifier. Let see how its intrinsic-gain (i.e., gain without load, a figure of merit) changes due to $W$ modifications at the same $g_m/I_d$ biasing. The small-signal model predicts that the intrinsec-gain

is equal to $g_m/g_{ds}$, and according with that Section 2 establishes, the $g_m/g_{ds}$ parameter is a *W*-invariant for the $g_m/I_d$ method.

We start the design using simulations with $W = 720$ nm (case 1 in Table 1).The size is then reduced by a factor of 3, two times, as well as the bias current, looking for a design with reduced current consumption without modifying the intrinsic-gain that is assumed to guaranteed by using the same $g_m/I_d$ biasing (hypothesis explained in Section 2). Nevertheless, please observe that in Table 1 the $g_m/I_d$ biasing remains quasi-invariant for the three cases, but the simulated intrinsic-gain not. It changed around 23% between cases 1 and 3. From the design point of view, it is not a slight variation.

Therefore, the intrinsic-gain is not *W*-invariant, and the $g_m/I_d$ method could conduct to an erroneous design using the UTBB FD-SOI 28 nm low-voltage-threshold technology.

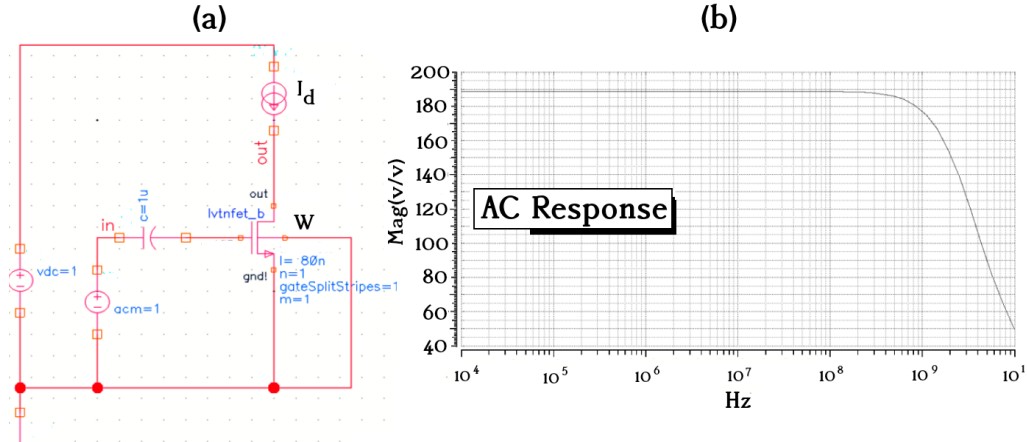

**Figure 5.** Case study: (**a**) common-source amplifier ($L = 80$ nm). The $I_d$ bias values and *W* sizes as indicated in Table 1, (**b**) simulated intrinsec-gain for the design case 1 (AC response). The theoretical intrinsec-gain is equal to $g_m/g_{ds}$.

**Table 1.** Case study: common-source amplifier, $L = 80$ nm, Figure 5.

| Parameters | Design 1 $W = 720$ nm $I_d = 450\,\mu\text{A}$ | Design 2 $W = 240$ nm (720/3 nm) $I_d = 150\,\mu\text{A}$ (450/3 $\mu$A) | Design 3 $W = 80$ nm (240/3 nm) $I_d = 50\,\mu\text{A}$ (150/3 $\mu$A) |
|---|---|---|---|
| $W$ (nm) | 720 | 240 | 80 |
| $I_d$ ($\mu$A) | 450 | 150 | 50 |
| $g_m$ (m$\Omega^{-1}$) | 1.06 | 0.34 | 0.11 |
| $\dfrac{g_m}{I_d}$ (V$^{-1}$) | 2.36 | 2.26 | 2.20 |
| $g_{ds}$ ($\mu\Omega^{-1}$) | 5.58 | 2.01 | 0.75 |
| $V_{th}$ (mV) | 354.92 | 352.31 | 348.02 |
| $G_V$ ($V/V$) (simulated intrinsec-gain) | 189.03 | 170.54 | 145.22 |
| $g_m/g_{ds}$ ($V/V$) | 189.96 | 169.15 | 146.17 |
| $C_{ds}$ ($aF$) | 6.61 | 2.37 | 0.93 |

## 7. Conclusions

This work showed numerically that the $g_m/I_d$ method requires more generality and systematization before being applied to UTBB FD-SOI 28 nm devices and more scaled technologies.



Regarding the UTBB FD-SOI understanding and usage advances for analog design, in particular, for high-frequency applications, the $g_m/I_d$ method should be handled carefully because the observed inaccurate predictions. These could be due to the extrinsic capacitances, parasitic resistances or side effects and these must be taken into account regarding the physical effects of fabrication process and in terms of the device's dynamic such as velocity saturation. There appears to be a large number of degrees of freedom and side effects in this technology that can not be ignored anymore (particularly finger partition and size), and the $g_m/I_d$ method (intended for hand analysis) requires much more generalization. This would lead the $g_m/I_d$ method unpractical or unmanageable so that advanced and optimal designs based on UTBB FD-SOI 28 nm low-voltage-threshold nfet must be performed by means of TCAD without assistance of reduced approaches like the $g_m/I_d$ method.

This study is the starting point to address more issues regarding the $g_m/I_d$ method due to these unexpected findings.

**Funding:** This research was funded by CSIC Grupos I+D 2018 Circuitos y Sistemas Integrados Biomédicos Autónomos y Conectados.

**Conflicts of Interest:** The authors declare no conflict of interest.

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
