# Peer review of "Evidence of Limitations of the Transconductance-to-Drain-Current Method (gm/Id) for Transistor Sizing in 28 nm UTBB FD-SOI Transistors"

_jlpea, doi:10.3390/jlpea10020017_

Round 1
Reviewer 1 Report
This paper presents limitations of gm/Id methodology in 28nm FDSOI process technology. However, this reviewer has three major concerns:
- It is written "Because the SOI transistor equations predict a bias current value that is proportional toW (ref. [10][11]), it is easy to conclude that all transistor cases are biased on the same gm/Id (see the detailed explanation in [5])". However, this argument needs to be validated because Vth of SOI transistor has a dependency on the width. The author should verify this argument based on PDK device model.
- For the simulation results, ss it post-layout simulation? If not, does the PCELL include intrinsic parasitic capacitances in the device model and automatically calculate them? In general, the degree of inclusion of parasitic capacitances in the schematic model may differ across the PDKs, so the author should clarify this before showing f_T results.
- The results provided in the manuscript is not enough to ensure the readers. Practical but simple design examples (i.e. inverter, CS amp, or diff amp) need to be added.
Author Response
Reviewer -1
------------
1- It is written "Because the SOI transistor equations predict a bias current value that is proportional to W (ref. [10][11]), it is easy to conclude that all transistor cases are biased on the same gm/Id (see the detailed explanation in [5])". However, this argument needs to be validated because Vth of SOI transistor has a dependency on the width. The author should verify this argument based on PDK device model.
------------
Answer
Thank you very much for identifying a weakness in our hypothesis. We checked the user reference: ‘‘UTSOI 2.1.0 User’s Manual CEA-LETI/MINATEC Campus, Grenoble, France, September 2015’’ (it is the PDK model). We confirm that the used model is not a Vth- based model and the Vth neither depends on W. This PDK model predicts the operating point parameters (among them the Vth voltage) using charge-surface potentials.
From the model of the used technology, we validate that Vth does not depend on W, but instead of L (expression 4.16 and 5.8 from UTSOI 2.1.0 User’s Manual). Unfortunately, we can not copy the model from the ‘‘UTSOI 2.1.0 User’s Manual’’ due to the signed ND- Agreement with the technology's vendor, that allows us to design using this kind of SOI transistors at university level.
However, we take into account the reviewer's suggestion, and to enhance the manuscript, we added text (with the main idea) in line 65 of the new manuscript version (red highlighted).
-----------
2 - For the simulation results, ss it post-layout simulation? If not, does the PCELL include intrinsic parasitic capacitance in the device model and automatically calculate them? In general, the degree of inclusion of parasitic capacitance in the schematic model may differ across the PDKs, so the author should clarify this before showing f_T results.
------------
Answer
Thank you very much for the observation that provides better manuscript completeness.
No, we didn't perform layout, then, ss (or thinking about the behavior of the inverse of gm/Id) is not a post-layout simulation. However, it is predicted using the complete PDK model on Spectre simulations.
On the other hand, the simulation includes intrinsic parasitic capacitances (the model includes them, such as the fringe capacitance as an example) but without adding extrinsic parasitic capacitances due to layout effects (with the extracted netlist from the cell).
From our point of view, it is enough to asses the theoretical gm/Id approach, in the sense that the mismatches ( between predictions using the method gm/Id and from full PDK model simulation ) arise without tanking into account extrinsic parasitic capacitances due to layout. It means that the core of the problem lies in the gm/Id approach and its hypothesis or instead of the PDK model. Extrinsic parasitic capacitances will turn the problem worse, but they are not the origin of the problem.
We take into account the reviewer's suggestion, and we explain in line 81-82 of the new manuscript version that the simulation uses intrinsic parasitic capacitances (red highlighted).
-----------
3 - The results provided in the manuscript is not enough to ensure the readers. Practical but simple design examples (i.e. inverter, CS amp, or diff amp) need to be added.
-----------
Answer
Thank you very much for the comments that improve the manuscript.
We added ''Section 6- Example: simple design case'' (red highlighted) . Here we explore the intrinsic-gain of a common-source amplifier. We show very briefly that moving the design requirements to low power consumption (reduced current bias), the predictions using gm over Id method does not hold, and circuit specifications change. The intrinsic-gain as W-invariant predicted by the technique is not maintained.
Reviewer 2 Report
Some minor spells or other minor details are commented on the attached pdf.
I find that the article could be of much more interest for the readers if the author could present a simple design case.

Author Response
Thank you very much for pointing out misspells. We corrected all the underlined in the file *.pdf you attached. Moreover, a carefully read has been performed to improve grammar. In addition to this, we added Section 6 – Simple design case. All changes are highlighted in red color.
Round 2
Reviewer 1 Report
I don't have any other comment.
Reviewer 2 Report
The author added a useful design case that increase the clarity of the paper. Only a short comment: should be "intrinsec" be written as "intrinsic"?